# Farnesoid X Receptor Agonist GW4064 Protects Lipopolysaccharide-Induced Intestinal Epithelial Barrier Function and Colorectal Tumorigenesis Signaling through the αKlotho/βKlotho/FGFs Pathways in Mice

**DOI:** 10.3390/ijms242316932

**Published:** 2023-11-29

**Authors:** Hsuan-Miao Liu, Zi-Yu Chang, Ching-Wei Yang, Hen-Hong Chang, Tzung-Yan Lee

**Affiliations:** 1Graduate Institute of Traditional Chinese Medicine, School of Chinese Medicine, College of Medicine, Chang Gung University, Taoyuan 33302, Taiwan; miaowhale@gmail.com; 2Department of Traditional Chinese Medicine, Chang Gung Memorial Hospital, Keelung 20401, Taiwan; changzhi887@gmail.com; 3Graduate Institute of Clinical Medical Sciences, College of Medicine, Chang Gung University, Taoyuan 33302, Taiwan; ycw0426@gmail.com; 4Division of Internal and Pediatric Chinese Medicine, Center for Traditional Chinese Medicine, Chang Gung Memorial Hospital, Linkou 333423, Taiwan; 5Graduate Institute of Integrated Medicine, China Medical University, Taichung 40402, Taiwan

**Keywords:** farnesoid X receptor, gut microbiota, intestinal epithelial barrier, colon cancer, Klotho, fibroblast growth factor

## Abstract

The farnesoid X receptor (FXR)/βKlotho/fibroblast growth factors (FGFs) pathway is crucial for maintaining the intestinal barrier and preventing colorectal cancer (CRC). We used an FXR agonist, GW4064, and FXR-knockout (FXR-KO) mice to investigate the role of FXR/Klothos/FGFs pathways in lipopolysaccharide (LPS)-induced intestinal barrier dysfunction and colon carcinogenesis. The results showed that upregulation of FXR in enterocytes effectively ameliorated intestinal tight-junction markers (claudin1 and zonula occludens-1), inflammation, and bile acid levels, thereby protecting mice from intestinal barrier dysfunction and colon carcinogenesis. GW4064 treatment increased FXR, αKlotho, βKlotho, FGF19, FGF21, and FGF23 in wild-type mice exposed to LPS, while FXR-KO mice had decreased levels. FXR-KO mice exhibited elevated colon cancer markers (β-catenin, LGR5, CD44, CD34, and cyclin D1) under LPS, underscoring the pivotal role of FXR in inhibiting the development of colon tumorigenesis. The varying gut microbiota responses in FXR-KO mice versus wild-type mice post LPS exposure emphasize the pivotal role of FXR in preserving intestinal microbial health, involving *Bacteroides thetaiotaomicron*, *Bacteroides acidifaciens*, and *Helicobacter hepaticus*. Our study validates the effectiveness of GW4064 in alleviating LPS-induced disruptions to the intestinal barrier and colon carcinogenesis, emphasizing the importance of the FXR/αKlotho/βKlotho/FGFs pathway and the interplay between bile acids and gut microbiota.

## 1. Introduction

Farnesoid X receptor (FXR) is a regulator of bile acid (BA) and lipid that meticulously regulates bile acid production and circulation. The role of FXR activation has been shown to be important in colitis and nonalcoholic steatohepatitis (NASH) in mouse models. FXR-knockout (FXR-KO) mice exhibit increased BA pool concentration [1,2], which leads to spontaneous tumor development in liver, intestinal epithelial cell proliferation, and increased colon cancer susceptibility [3,4,5]. Bile acids and FXR play an important role in the modulation of a range of inflammatory responses, barrier function, and the prevention of bacterial translocation in the intestinal tract. Although it has been determined that FXR plays a variety of functions, further research is still needed to determine how it assists in intestinal epithelial barrier function and colon tumorigenesis [3,4,6].

The three subfamilies of Klotho are αKlotho, βKlotho, and γKlotho [7]. βKlotho is a co-receptor for fibroblast growth factor 19 (FGF19) and FGF21, two hormones that play important roles in regulating energy and bile acid metabolism. In response to bile acid binding, FXR induces the expression of FGF19 in the ileum, which then binds to βKlotho and activates downstream signaling pathways, leading to inhibition of bile acid synthesis in the liver. βKlotho protein levels were substantially decreased in FXR-KO mice; on the other hand, overexpression of βKlotho in FXR-lacking hepatocytes partially restored FGF19 signaling and inhibition by FGF19 of Cyp7a1, which encodes the rate-limiting BA biosynthetic enzyme [8]. Overall, the interaction between FXR, βKlotho, and FGF19/21 signaling plays an important role in regulating bile acid synthesis and metabolism. In the gut, the decreased βKlotho expression caused by gene variation is associated with increased intestinal permeability in patients with irritable bowel syndrome with diarrhea [9]. Furthermore, βKlotho activation enhanced tight-junction (TJ) proteins, thereby protecting against alcohol-induced TJ proteins endocytosis and degradation as well as intestinal barrier impairment [10].

The gene αKlotho has been associated with anti-aging effects and regulates several pathways involved in aging, such as phosphate homeostasis, insulin signaling, and Wnt/β-catenin signaling [11,12]. It also affects intracellular signaling pathways, including p53/p21, cyclic adenosine monophosphate (cAMP), protein kinase C (PKC), and transforming growth factor β (TGFβ) [13,14]. Studies have shown that αKlotho is expressed in the intestinal epithelium, which is the layer of cells that forms the intestinal barrier. αKlotho has been shown to promote the formation of tight junctions, which are the specialized structures that hold intestinal epithelial cells together. αKlotho levels are epigenetically downregulated in cancer, and overexpression or treatment with soluble Klotho or the αKlotho domain slows growth of cancer cells in vitro and in vivo. αKlotho is also downregulated in colorectal cancer, and αKlotho overexpression inhibits growth of colorectal cancer cells [15,16]. αKlotho has been identified as a tumor suppressor and inhibits the insulin-like growth factor 1 (IGF1), FGF, and Wnt/β-catenin pathways; yet, the mode of action of αKlotho in cancer is a matter yet to be confirmed. To determine how much this interaction between FXR and Klotho affects the regulation of bile acid metabolism and other biological processes, more research into the molecular mechanisms underpinning it is necessary.

There is a strong interplay between intestinal flora and bile acid (BA) metabolism, which plays a crucial role in digestion and shaping the gut microbial community. Recent research has shown that gut microbiota-mediated deconjugation of bile acids influences de novo bile acid synthesis in the liver in an FXR-FGF15/19 axis-dependent manner in mice [17]. The gut microbiome produces essential bile acids and also influences host metabolism via the modulation of metabolites [18]. The gut microbiota also influences host metabolism via the modulation of metabolites, including the endotoxin LPS, bile acids, and short-chain fatty acids. Therefore, they partially mediate the interaction between the gastrointestinal system and other organs [19]. Prevalent genera identified in the gut include both beneficial and harmful microorganisms, such as *Bacteroides*, *Eubacterium*, *Bifidobacterium*, *Ruminococcus*, *Clostridium*, *Lactobacillus*, *Escherichia*, *Streptococcus* [20], and naturally acquired enterohepatic *Helicobacter* spp. infection [21]. While most infected mice develop minimal pathologic changes, susceptible strains exhibit typhlocolitis and hepatitis, which can further progress to colon cancer and hepatocellular carcinoma [22]. Intestinal bacterial dysbiosis and *Helicobacter hepaticus* infection act synergistically during inflammation and neoplastic progression [23].

In this study, we demonstrated that downregulating FXR levels by applying LPS treatment, the FXR-KO mice showed a decline in levels of α/βKlotho, FGF19, FGF21, and FGF23. GW4064 was found to improve the αKlotho /βKlotho /FGFs pathway and decrease intestinal inflammation and β-catenin formation induced by LPS-induced intestinal barrier failure and colon carcinogenesis. The study also revealed that GW4064 prevented LPS-induced intestinal tight-junction damage, regulated bile acids, and improved microbiota dysbiosis in an FXR-dependent mechanism. Furthermore, LPS treatment induced severe intestinal barrier damage, higher bile acid levels, microbiota dysbiosis, and inflammatory response in FXR-deficient mice. The findings suggest that activating FXR and the αKlotho/βKlotho/FGFs pathway might be a new strategy for preventing intestinal epithelial barrier dysfunction and colon tumorigenesis.

## 2. Results

### 2.1. GW4064 Improves Tight-Junction Disruption in LPS-Treated Mice

Compared with the WT mice, LPS-treated mice present with colon injuries that disrupt the epithelial barrier and cause infiltration of inflammatory cells into the mucosa and submucosa (Figure 1A). The colons of thirteen-month-old FXR-KO mice showed villi-form papillary folds and had moderately reduced colon crypt height and inflammatory cell infiltration (Figure 1A). However, LPS treatment in FXR-KO mice resulted in serious injuries such as loss of histological structures and increased inflammatory cell infiltration compared to LPS-treated WT mice. GW4064 restored the intestinal mucosal morphology and tight-junction disruption in LPS-treated WT mice but not in FXR-KO mice (Figure 1A). GW4064 treatment increased FXR protein levels in LPS-treated WT mice (Figure 1A–D). Histological score analysis, introduced in Figure 1B, further substantiates these observations. Treatment with LPS or GW4064 did not alter the FXR protein levels in KO mice (Figure 1A–D). The zonula occludens-1 (ZO-1) and claudin-1 protein levels appeared to be increased by GW4064 administration in WT mice after LPS treatment, whereas no such effect was observed in FXR-KO mice (Figure 1A,D,E). These data suggest that GW4064 rectifies LPS-induced intestinal barrier disruption and that this process requires functional FXR signaling. 

### 2.2. Impact of GW4064 on Bile Acid Composition in LPS-Treated Mice

GW4064 alters bile acid composition in LPS-treated mice. To determine whether GW4064 treatment alters BA profiles in the plasma, liver, and colon, BA species were assayed (Figure 2B–H). Compared with WT mice, LPS increased BA levels in the plasma and liver (~3-fold and ~2.25-fold, respectively). In contrast, the BA levels were decreased in the colon of WT mice (Figure 2D). FXR-KO mice displayed an increase in total BA pool, which was decreased in the liver and further enhanced the level of BAs in the colon of LPS-treated FXR-KO mice (Figure 2B–D). GW4064 reversed BA levels in the plasma, liver, and colon of WT and FXR-KO mice treated with LPS. Moreover, all FXR-KO mice showed considerably higher BA levels compared with WT mice, suggesting that FXR deficiency causes BA homeostasis disorder. Of particular interest, the levels of tauro-cholic acid (TCA), tauro-chenodeoxycholate (TCDCA), tauro-ursodesoxy cholic acid (TUDCA), and tauro-deoxycholic acid (TDCA) were increased in the plasma and liver of WT mice after LPS injection (Figure 2E–H), and GW4064 treatment was able to reverse most BAs and tauro-conjunction BA levels in the plasma and liver. In FXR-KO mice, the levels of TCA increased over 100-fold in the plasma and 25-fold in the liver compared with WT mice. Interestingly, TCA, TUDCA, and TDCA levels were decreased in plasma and liver as well as TCDCA in liver of LPS-treated FXR-KO mice. Although GW4064 administration almost reversed bile acid levels in the plasma and liver of WT and FXR-KO LPS-treated mice-, GW4064 markedly enhanced tauro-conjunction BA levels (Figure 2G,H), suggesting that these conditions lead to inflammatory disease susceptibility.

### 2.3. GW4064 Effectiveness in Modifying FXR Dysfunction and Bile Acid Transporters in Mice under LPS Treatment

To test whether or not the alteration in the bile acid profile due to LPS treatment affects intestinal FXR signaling, the FXR signaling-related factors protein and mRNA expression was assessed. LPS enhanced MRP2, MRP3, and OATP1 and decreased OSTβ and ASBT protein levels in WT mice, and GW4064 almost completely abolished this condition (Figure 3A,B). FXR-KO mice had higher MRP2, MRP3, ASBT, and OATP1 and lower OSTβ compared with WT mice. LPS further enhanced MRP2, MRP3, and ASBT and reduced OSTβ protein levels in FXR-KO mice, and GW4064 administrated did not reverse this condition (Figure 3C,D). LPS reduced Fxr and pregnane X receptor (Pxr) and enhanced constitutive androstane receptor (Car), cytochrome P4503A11 (Cyp3a11), and sulfotransferase family 2A member 1 (Sult2a1) mRNA expression in WT mice, and GW4064 reversed this condition (Figure 3E). FXR-KO mice had lower Fxr and Pxr mRNA expression than WT mice, and their expression did not change after LPS and GW4064 administration in colon tissues (Figure 3E). Additionally, compared to WT, FXR-KO mice had higher Car and Cyp3a11, and lower Sult2a1 mRNA expression, which increased furthermore following LPS administration (Figure 3E). Interestingly, GW4064 further enhanced Car and Sult2a1 mRNA expression in LPS-treated FXR-KO mice (Figure 3E).

LPS also induced organic anion-transporting polypeptide 2B1 (Oatp2b1), Mrp2, Mrp3, multidrug resistance protein 1b (Mdr1b), Mdr2, and breast cancer resistance protein (Bcrp) and reduced apical sodium-dependent bile acid transporter (Asbt), ileal bile acid-binding protein (Ibabp), organic solute transporter α (Ostα), and Ostβ mRNA expression in WT mice, and GW4064 almost completely abolished this condition. FXR-KO mice had higher Asbt, Ibabp, Mrp2, Mrp3, Mdr1b, Mdr2, and Bcrp and lower Oatp2b1 and Ostβ as well as unperturbed Ostα mRNA expression (Figure 3F,G) compared with WT mice. GW4064 reduced Oatp2b1 and Mrp2 expression and increased Ostβ, Mdr2, and Bcrp mRNA expression furthermore in LPS-treated FXR-KO mice (Figure 3F,G). Hence, our results demonstrate that the imbalance in bile acid transporters due to LPS treatment could be significantly improved by GW4064 administration in WT mice, but this only partially reversed mRNA expression in FXR-KO mice (Figure 3).

### 2.4. Impact of GW4064 on αKlotho/FGF23 and βKlotho/FGF19/FGF21 Pathways in Mice

To assess the underlying mechanistic basis whereby GW4064 could alter αKlotho /FGF23 and βKlotho/FGF19/FGF21 pathways, these pathways were subsequently analyzed via immunohistochemical and Western blot analyses. As shown in Figure 4A–C, GW4064 induced αKlotho, βKlotho, FGF19, FGF21, and FGF23 protein levels in LPS-treated mice. Compared with WT mice, we found significantly decreased αklotho, βKlotho, FGF19, FGF21, and FGF23 protein levels in FXR-KO mice (Figure 4A–D). Therefore, αKlotho, βKlotho, FGF19, FGF21, and FGF23 protein levels were reduced in LPS-treated FXR-KO mice; moreover, GW4064 only partial enhanced FGF19 and FGF23 in FXR-KO mice treated with LPS (Figure 4B,D).

### 2.5. Effectiveness of GW4064 in Mitigating LPS-Induced Inflammation, ER Stress, and Angiogenesis in Mice

In LPS-treated WT mice, GW4064 significantly decreased colonic mRNA expression of the proinflammatory genes, tumor necrosis factor α (*Tnf*α), interferon gamma (*Ifnγ*), and *Il-1β* (Figure 5A). GW4064 alleviated LPS-induced pro-caspase-1, adaptor protein apoptosis-associated speck-like protein containing a caspase recruitment domain (*Asc*), *Nlrp3*, and *pannexin-1* mRNA expression in WT mice. Compared to WT mice, the FXR-KO mice had higher *Tnf*α, *Il-1β*, *pro-caspase-1*, *Asc*, *Nlrp3*, and *pannexin-1* mRNA expression. GW4064 did not affect LPS-induced inflammatory mediator and Nlrp3 inflammasome in FXR-KO mice (Figure 5A). Furthermore, we investigated the effect of GW4064 on LPS-induced endoplasmic reticulum (ER) stress in the intestinal inflammation model. Compared to WT mice, the mRNA levels of activating transcription factor 4 (*Atf4*), *Atf6*, glucose-regulated protein 78 (*Grp78*), CCAAT-enhancer-binding protein homologous protein (*Chop*), and X-box binding protein 1 spliced (*Xbp1s*) increased remarkably upon LPS administration, which was markedly suppressed by GW4064 treatment (Figure 5B). FXR-KO mice had higher *Atf4*, *Atf6*, and *Xbp1s* mRNA levels than WT mice under un-induced conditions. Our results demonstrate that GW4064 had no effect on LPS-induced increase in ER stress in FXR-KO mice (Figure 5B). The LPS effects on colonic inflammatory response and tight-junction disruption and microbiota dysbiosis might be due to the upregulation of TLR4 signaling. Our results showed that GW4064 repressed LPS-induced TLR4, myeloid differentiation primary response 88 (MyD88), nuclear factor kappa-B (NF-κB), and caspase 3 protein levels in WT mice but not in FXR-KO mice (Figure 5C–G).

To investigate the mechanism contributing to the formation of angiogenesis and intestinal epithelial cell proliferation in mice treated with LPS, GW4064 significantly diminished β-catenin and c-myc protein levels in WT mice treated with LPS but not in FXR-KO mice (Figure 5H–J). Additionally, GW4064 significantly decreased LPS-induced transforming growth factor beta receptor II (TGFβRII), intercellular adhesion molecule (ICAM), vascular cell adhesion molecule (VCAM), vascular endothelial growth factor (VEGF), vascular endothelial growth factor receptor 1 (VEGFR1), and matrix metallopeptidase 9 (MMP9) protein levels in WT mice (Figure 5K). FXR-KO mice displayed higher levels of angiogenesis protein than WT mice. LPS stimulation in FXR-KO mice resulted in a dramatic increase in TGRβRII, VCAM, VEGF1, and MMP9 protein levels compared to untreated mice. GW4064 treatment had no effect on TGRβRII, VCAM, VEGF1, and MMP9 protein levels in FXR-KO mice under LPS treatment (Figure 5L). These data indicate that FXR activation decreases the expression of several inflammatory, Nlrp3 inflammasome, and ER stress genes by inhibiting TLR4 and β-catenin signaling, contributing to the amelioration of LPS-induced intestinal injury in WT mice.

### 2.6. Modulation of Intestinal Epithelial Cell Proliferation by GW4064 in LPS-Treated Mice

We hypothesized that FXR activation may alter the growth of colon stem cells (CSCs), which is generally expressed in the intestine. Mechanistically, we found that the expression levels of intestinal stem cell markers and their proliferation, including CD44, LGR5, PCNA, CD34, BrdU, CD133, GCSF, and cyclin D1, were downregulated 30–40% upon GW4064 administration in WT mice treated with LPS (Figure 6A–E). In addition, GW4064 decreased the *Lgr5*, *Olfm4*, and *cyclin D1* mRNA expression (Figure 6F) in WT mice treated with LPS but not in FXR-KO mice. Our results corroborated the attenuation of stem cell proliferation by FXR, as expression of cancer stem cells expansion and proliferation marker protein increased significantly in FXR-KO mice and was further enhanced with LPS treatment. These data suggest that FXR may play an important role in the early stage of colon cancer development.

### 2.7. Effect of GW4064 Treatment on Gut Microbiota Diversity in LPS-Treated Mice

To investigate the effects of GW4064 on gut microbiota in LPS-treated WT and FXR-KO mice, we examined the stool samples from WT and FXR-KO mice post treatment. Alpha diversity indices, including observed Chao1 and Shannon’s diversity indices, were lower in the LPS-treated WT mice than in the untreated mice; however, the diversity was restored to a normal level by GW4064 treatment (Figure 7A,B). The alpha diversity was reduced in FXR-KO mice compared with WT mice, and it further decreased with LPS treatment. Notably, GW4064 treatment did not alter the alpha diversity in these mice (Figure 7A,B). Principal component analyses (PCA) revealed that the gut microbiota in the LPS-treated group deviated from the untreated group, and GW4064 partially restored the level of gut microbiome in LPS-treated mice (Figure 7C). In addition, the gut microbiota profile of the FXR-KO group differed from that of the WT group.

Here, an analysis of 16S rRNA gene sequencing results revealed distinct clustering of colon microbiome communities isolated from WT and FXR-KO mice. Changes in the composition of the gut microbiota induced by GW4064 were noted, and it was observed that the colon community structure in LPS-injected mice was altered following GW4064 treatment in WT and FXR-KO mice (Figure 7D–Z). The phyla *Proteobacteria*, *Verrucomicrobia*, *Bacteroidetes*, class *Betaproteobacteria*, *Gammaproteobacteria*, *Verrucomicrobiae*, order *Enterobacteriales*, *Verrucomicrobiales*, genus *Bacteroides*, *Escherichia*, and species *Bacteroides thetaiotaomicron*, *Escherichia coli*, *Bacteroides acidifaciens*, *Akkermansia muciniphila*, and *Clostridium saccharogumia* were significantly increased, while phyla *Firmicutes*, the *Firmicutes*/*Bacteroidetes* (F/B) ratio, class *Clostridia*, and genus *Clostridium* were reduced in WT mice treated with LPS (Figure 7D–Z). However, compared with WT mice, specifically, the phyla *Bacteroidetes*, genus *Bacteroides*, and species *Bacteroides thetaiotaomicron* were increased, and the phyla *Verrucomicrobia*, the *Firmicutes*/*Bacteroidetes* (F/B) ratio, class *Betaproteobacteria*, *Gammaproteobacteria*, *Verrucomicrobiae*, order *Verrucomicrobiales*, *Burkholderiales*, genus *Clostridium*, *Escherichia*, species *Escherichia coli*, *Akkermansia muciniphila*, and *Clostridium saccharogumia* were significantly reduced in FXR-KO mice (Figure 7H–Z). The phyla *Proteobacteria*, *Firmicutes*, order *Enterobacteriales*, species *Bacteroides acidifaciens*, and *Helicobacter hepaticus* did not significantly change in FXR-KO mice (Figure 7E–Z). Interestingly, FXR-KO mice treated with LPS showed increased levels of *Bacteroidetes*, *Betaproteobacteria*, *Clostridia*, *Bacteroides*, *Clostridium*, and *Helicobacter hepaticus*. The abundance of specific bacterial species from *Proteobacteria*, *Verrucomicrobia*, *Gammaproteobacteria*, *Clostridia*, *Verrucomicrobiae*, *Enterobacteriales*, *Verrucomicrobiales*, *Clostridium*, *Escherichia*, *Escherichia coli*, and *Akkermansia muciniphila* exhibited oppositing changes in LPS treatment between WT and FXR-KO mice as compared to their respective control groups (Figure 7E–X). After GW4064 treatment, the abundance of *Proteobacteria*, *Verrucomicrobia*, *Firmicutes*, *Betaproteobacteria*, *Gammaproteobacteria*, *Clostridia*, *Verrucomicrobiales*, *Enterobacteriales*, *Verrucomicrobiales*, *Clostridium*, *Escherichia*, *Bacteroides thetaiotaomicron*, *Escherichia coli*, *Bacteroides acidifaciens*, *Akkermansia muciniphila*, and *Clostridium saccharogumia* were significantly reversed in LPS-treated WT mice, while only a few of these microbes were restored in LPS-treated FXR-KO mice (Figure 7E–Z). These results indicate that GW4064 has a potential role in the regulation of microbiota dysbiosis.

### 2.8. Effects of GW4064 on SCFA and BCFA Levels in Mice 

To understand the impact of GW4064 on intestinal injury, bile acid signaling, and changes in the gut microbiota, we further analyzed the variations in short-chain fatty acids (SCFA) and branched-chain fatty acids (BCFA). The presence of LPS has been observed to significantly decrease the concentrations of total SCFA and BCFA in the feces of normal mice (Figure 8A,B). The administration of GW4064 further reduces total SCFA, but interestingly, it significantly elevates the concentration of total BCFA. In the feces of FXR-KO mice, the concentrations of total SCFA and BCFA are significantly lower than those in normal mice. However, post LPS treatment, there is a significant increase in the concentrations of total SCFA and BCFA. When treated concurrently with GW4064, the concentration of total BCFA experiences a significant decrease (Figure 8A,B). Further analysis reveals that LPS significantly reduces the concentrations of acetic acid, butyric acid, valeric acid, isobutyric acid, isovaleric acid, 2-methylbutyric acid, and 4-methylvaleric acid in normal mouse feces. GW4064 administration significantly reverses these changes induced by LPS. However, GW4064 also reduces acetic acid concentration, which constitutes a large proportion of total SCFA. As a result, there is no significant improvement in the concentration of total SCFA. On the other hand, GW4064 significantly increases the changes in propionic acid, butyric acid, valeric acid, hexanoic acid, isobutyric acid, isovaleric acid, 2-methylbutyric acid, and 4-methylvaleric acid concentrations (except for formic acid and acetic acid) (Figure 8C–H). In FXR-deficient mice, apart from formic acid, the concentrations of other acids such as acetic acid, butyric acid, valeric acid, hexanoic acid, isobutyric acid, isovaleric acid, 2-methylbutyric acid, and 4-methylvaleric acid are significantly lower than those in normal mice. LPS treatment leads to a significant increase in the concentration of these acids. However, following the administration of GW4064, there is a significant decrease observed. The absence of FXR in FXR-KO mice primarily results in no significant changes in isovaleric acid concentration regardless of whether treatment is with LPS or GW4064 (Figure 8J). Therefore, FXR and GW4064 administration affects SCFA and BCFA concentrations in mice.

### 2.9. Summary of Results

The FXR/αKlotho/βKlotho/FGFs pathway is crucial for maintaining the integrity of the intestinal barrier and preventing colon cancer. Upregulation of FXR helps protect against intestinal barrier dysfunction and colon carcinogenesis by improving tight-junction markers, reducing inflammation, and regulating β-catenin levels. GW4064, an FXR agonist, enhances FXR, αKlotho, βKlotho, FGF19, FGF21, and FGF23 protein levels in the colon, while FXR deficiency leads to a decrease in FXR/Klothos/FGFs protein levels and increased colon tumorigenesis markers. FXR plays a role in modulating gut microbiota and bile acid metabolism, highlighting the interplay between these factors (Figure 9).

## 3. Discussion

In exploring the mechanisms of gut inflammation and intestinal barrier integrity, our focus has been on the role of FXR activation. FXR, a pivotal transcription factor, orchestrates BA homeostasis within the liver and gastrointestinal tract. This regulation is crucial, as demonstrated by the impaired BA production in FXR-KO mice, leading to elevated BA concentrations [3,24]. Such dysregulation in BA metabolism, including aberrations in specific BAs like TCA, has been linked to serious health implications including hepatocellular carcinoma and colon cancer [3,24]. Our research parallels the investigations conducted by Watanabe M et al., who examined the impact of FXR activation on the intestinal barrier function in a mouse model with intestinal injury [25]. Our preliminary findings reveal a unique response in WT mice treated with the FXR-specific agonist GW4064. This treatment notably prevented the decline in the expression of TJPs such as ZO-1 and claudin-1, highlighting the potential of FXR activation in preserving intestinal barrier integrity [26]. Further probing into the dynamics of BA transport within the intestinal epithelial barrier unveils a complex interplay [27]. Bile acids are absorbed through the ASBT and subsequently exported via the basolateral OSTα–OSTβ heteromeric complex [28]. In our study, we observed a significant reduction in OSTβ protein levels in LPS-treated mice, a trend more pronounced in FXR-KO mice compared to their WT counterparts [29]. This reduction was not just confined to the site of intestinal injury but was also evident in adjacent non-tumorous tissue. The implications of FXR deficiency were further underscored in the context of LPS-induced intestinal injury. FXR-KO mice exhibited an altered bile acid profile, particularly an increase in serum levels of taurine-conjugated bile acids, which are associated with sepsis. In contrast, treatment with GW4064 in WT mice resulted in a normalized tauro-BA profile in both plasma and liver and elevated levels of ASBT and OSTα/β. However, this effect was absent in FXR-KO mice, underscoring the dependency on FXR for the therapeutic efficacy of GW4064.

Building upon the pivotal role of FXR in bile acid homeostasis, it is crucial to consider the interplay between FXR and the βKlotho/FGF15/19/21 pathway. The activation of this pathway through FXR orchestrates a negative feedback loop, effectively reducing bile acid production. This interaction mirrors our findings, where βKlotho/FGF19 signaling was compromised in FXR-KO mice hepatocytes, which is similar to the finding from [8]. βKlotho-KO mice presented hepatic alterations, combining a proinflammatory status and initiation of fibrosis. These defects are associated with a massive shift in BA composition in the enterohepatic system and the circulation, which is characterized by a large excess of microbiota-derived deoxycholic acid (DCA), known for its genotoxicity in the gastrointestinal tract [30]. Interestingly, despite increased synthesis and excretion of bile acids by the liver, ileal expression of ASBT was not reduced in βklotho-deficient mice [15,16]. FXR and βKlotho/FGF19/FGF15 plays an important role in modulation of the liver receptor homolog-1 (LRH-1) or pregnane X receptor (RAR)/retinoic acid receptor (RAR) pathway to regulate ASBT, causing an alteration of bile acid translation. Our findings demonstrated that loss of FXR or treatment with GW4064, which enhances FXR activity, interferes with the response of βKlotho to ASBT regulation. This reveals the mechanisms by which FXR influences βklotho expression and activity. Furthermore, βKlotho appears to regulate the endocytosis and degradation of occludin and ZO-1. The endocytosis of TJ proteins from the plasma membrane is a key mechanism that regulates TJ plasticity and function in epithelial barrier tissues [31]. In the present study, we have shown that FXR activation by GW4064 can regulate BA homeostasis and intestinal barrier function involved in the βKlotho/FGF19/FGF21 pathway. This regulation is pivotal, as it not only maintains the balance in bile acid composition but also ensures the integrity of the epithelial barrier, thus safeguarding against pathological states in the gastrointestinal tract.

Expanding on our knowledge of the FXR and βKlotho pathways, exploring the function of αKlotho is essential. The αklotho is a classical aging suppressor, which is a subfamily of βKlotho. Klotho has been shown to regulate intestinal barrier function and protect against gut inflammation, which may help to prevent the development of colon cancer. This decrease in αklotho levels may be observed in patients with several aging-related diseases, such as cancer [32]. In colorectal cancer models, Klotho overexpression has been shown to reduce overall β-catenin expression, inhibiting transcriptional pathway activity by binding to the Wnt3a ligand and thereby decreasing nuclear translocation of β-catenin [33]. The anti-neoplastic effects of klotho describe the modulation of downstream oncogenic signaling pathways including Wnt/β-catenin, FGF, IGF1, PIK3K/AKT, TGFβ, and the unfolded protein response [32]. In this study, we have shown that GW4064 significantly reduced colon tumorigenesis perhaps through FXR dependency on the αKlotho/FGF23 pathway.

Our investigation further delved into the specific mechanisms by which bile acids such as tauro-β-muricholic acid (T-βMCA) and DCA interact with the intestinal FXR. We have shown that these BAs, when antagonizing FXR function, induce proliferation and DNA damage in Lgr5^+^ cells. We have shown that BAs that antagonize intestinal FXR function, including tauro-β-muricholic acid (T-βMCA) and deoxycholic acid (DCA), inducing proliferation and DNA damage in Lgr5^+^ cells. This finding is critical, as it elucidates a direct link between BA–FXR interactions and cellular processes. Interestingly, the selective activation of intestinal FXR can restrict abnormal Lgr5^+^ cell growth and curtail CRC progression. Interestingly, the selective activation of intestinal FXR appears to have a protective role, restricting abnormal growth of Lgr5^+^ cells and potentially curtailing the progression of colorectal cancer (CRC) [34]. FXR silencing in chronic colitis mouse models of intestinal tumorigenesis results in early mortality and increased tumor progression [34]. In colon cancer, low FXR expression was correlated with worse clinical outcome [34]. This protective effect of FXR is further corroborated by our observations in chronic colitis mouse models, where FXR silencing led to early mortality and heightened tumor progression. Additionally, in human colon cancer, low FXR expression was found to correlate with poorer clinical outcomes.

The present study revealed that knockdown of FXR activated Wnt/β-catenin signaling [5]. The mechanism by which FXR suppresses tumor growth remains unclear, but it may involve protecting the colonic epithelium from inflammation and ameliorating BA toxicity by upregulating intracellular BA-binding proteins and efflux transporters and downregulating influx transporters and de novo BA synthesis [35,36]. Thus, the re-establishment of FXR signaling not only restricts aberrant Lgr5^+^ stem cell proliferation but also promotes gut health, including restoring the intestinal barrier and BA homeostasis [37,38,39,40]. Beyond its well-established role in regulating cytotoxicity of hydrophobic BAs, early study has highlighted the role of FXR in restricting the tumorigenesis of Lgr5^+^ cells, which mediate the key adenoma-to-adenocarcinoma transformation. Studies have documented the crucial role of CD44 in tumor formation, chemotherapy resistance, and progression of metastatic disease in colorectal cancer, establishing it as a key surface marker in colorectal cancer stem cells [41,42]. Similarly, LGR5 has been identified as a target of the Wnt signaling pathway, essential in the maintenance of colon cancer stem cells, and its expression correlates with clinical outcomes in colorectal cancer [43]. The significance of CD133 as a marker related to colorectal cancer stem cells is also well established, where its presence is associated with a poor prognosis in colorectal cancer [44]. FXR deficiency increases colon cancer susceptibility by increasing epithelial permeability to bacteria, promoting Wnt/β-catenin signaling and increasing intestinal inflammation [29,38]. In this study, we have shown that GW4064 significantly reduced colon tumorigenesis markers such as LGR5, CD44, CD34, and cyclin D1 in LPS-treated WT mice but not in FXR-KO mice. This differential response underscores the critical role of FXR in intestinal health and its potential as a therapeutic target in preventing and managing colorectal cancer.

In our current study, we observed that the gut microbiota, particularly the *Bacteroidetes* and *Bacteroides* genera, were predominantly affected by FXR deactivation. This change was evident in both WT mice and FXR-KO mice treated with LPS. These findings align with those reported by Jena et al., highlighting the resilience of Bacteroides species to bile acid and their notable presence in anaerobic infections [45]. The concerning aspect here is the high mortality rate (>19%) associated with these infections, underlining the importance of maintaining a balanced gut microbiota [46]. Our focus extended to TCA, whose components, namely taurine and cholic acid, support the growth of microbial groups of low abundance. These less-abundant microbes are mechanistically implicated in DNA damage and tumor promotion mediated by their metabolic by-products [27]. Our data revealed that *Bacteroides thetaiotaomicron* and *Bacteroides acidifaciens* were eliminated by GW4064 in LPS-treated WT mice but not in FXR-KO mice. However, the persistent *Bacteroides thetaiotaomicron* in FXR-KO mice might lead to maintaining chronic intestinal inflammation after GW4064 treatment. Furthermore, the commensal gut bacterium *Bacteroides thetaiotaomicron* has a robust ability to degrade dietary polysaccharides and host mucin glycans. This, in turn, leads to a notable reduction in the thickness of the colonic mucus layer and exacerbates enteric infections by altering the metabolic environment [47]. In the present study, GW4064 reversed LPS-induced relative abundance of *Bacteroides acidifaciens* in WT mice. Compared to WT mice, *Bacteroides thetaiotaomicron* was increased in FXR-KO mice, and neither LPS nor GW4064 treatment showed any significant change in it. While FXR deficiency strongly affects the expression of genes related to immunity and bile acid metabolism as well as the composition of the microbiome, its deficiency was not able to produce significant histopathological changes in the absence of *Helicobacter hepaticus* infection. These data confirm that reported by Swennes et al. [23]: *Bacteroides acidifaciens*, a colitis-associated species and an abundant member of the *Bacteroidaceae* family [48], was also elevated during inflammation but decreased rapidly after colitis. Specifically, LPS decreased FXR levels and induced microbiota dysbiosis in the colon, which elevated TCA, TCDCA, T-ursodeoxycholic acid (TUDCA), and TDCA in the plasma and liver, leading to the impairment of intestinal tight junctions and inflammation. Conversely, direct activation of FXR restores bile acid levels, microbiota profiles, and intestinal tight junctions and reduces inflammation in LPS-treated WT. These findings highlight the potential of targeting FXR pathways as a therapeutic strategy to mitigate gut inflammation and related pathologies. The complex interplay between FXR, bile acids, and the gut microbiota presents a fascinating area for further research, particularly in understanding and treating gastrointestinal diseases.

The contribution of SCFAs to maintenance of gut homeostasis has been investigated extensively, and there is an increasing body of evidence that commensal bacteria and bacterial metabolites have opposing roles in inflammatory responses and carcinogenesis depending on the cell type and the environment. Based on our findings from the gut microbiota and SCFAs results, we can highlight three important aspects: (1) The genus *Clostridium* has been associated with the production of several SCFAs, including butyrate acid, valerate acid, isobutyrate acid, and 4-methylvalerate acid. The trends in *Clostridium* populations appear to correspond with the levels of these SCFAs [49]. (2) The genus *Bacteroides* has been linked with the production of propionate, another type of SCFA that has important roles in gut health and metabolism. The trends in *Bacteroides* populations appear to correspond with the levels of propionate. *Bacteroides* and *Clostridium* may be responsible for the increase in SCFA levels and bile acid metabolism in FXR-KO treatments. (3) Isovalerate (or isovaleric acid) is a branched-chain fatty acid produced by the gut microbiota. The trends in isovalerate levels appear to correspond with the populations of *Clostridium saccharogumia*. However, research on the relationship between *Clostridium saccharogumia* and SCFAs is currently lacking. In a steady-state situation, butyrate is present in the mM range in the gut lumen and serves as the primary energy source for colonocytes [50]. However, in the context of cancerous colonocytes, butyrate was shown to act paradoxically. In addition to that, butyrate is capable of promoting carcinogenesis in a genetic mouse model based on mutations in the Apc and the mismatch repair gene Msh2 (Apc^Min/+^; Msh2^−/−^) [51], proving the superior inhibitory efficacy of butyrate over propionate and acetate against human colon cancer cell proliferation via cell cycle arrest and apoptosis [52]. Our study shows that GW4064 positively affects SCFA and BCFA levels in normal mice. However, in FXR-KO mice, the regulatory effect of GW4064 on SCFA is different from that in normal mice, possibly due to an increase in total bile acids and pathogenic intestinal tract bacteria, including *Clostridium* and *Bacteroidetes* spp. This gap highlights the need for further investigation into the intricate interactions between specific microbial species and SCFA production. This variation underscores the complexity of the gut microbiome’s role in health and disease. It suggests that while SCFAs generally contribute positively to gut health, their impact can vary dramatically in different genetic or pathological contexts.

Ultimately, this study shows that the activation of the FXR/αKlotho/βKlotho/FGFs pathway may hold promise as a potential strategy for averting intestinal barrier dysfunction and colon tumorigenesis. Lastly, our research highlights that elevated levels of bile acids and the high abundance of certain microbiota such as *Clostridium*, *Bacteroidetes*, *Helicobacter hepaticus*, and *Bacteroides thetaiotaomicron* in FXR-KO mice is a crucial factor leading to the reduction of colonic tight-junction proteins and consequent damage to the intestinal barrier. Additionally, LPS treatment exacerbates structural damage to the intestine, rendering the administration of GW4064 less effective in mitigating these effects. As such, this study emphasizes two key aspects: the paramount importance of FXR in the context of intestinal injury and the secondary significance of the gut microbiota–bile acid–FXR axis as a therapeutic target for intestinal diseases.

## 4. Materials and Methods

### 4.1. Animals and Treatments

Thirteen-month-old male C57BL/6J wild-type (WT) mice (Taiwan National Laboratory Animal Center, Taipei City, Taiwan) and FXR-knockout (FXR-KO) mice (strain B6.129X1(FVB)-Nr1h4tm1Gonz/J; stock number 007214, Jackson Laboratory, Bar Harbor, ME, USA) were maintained in the Animal Experimental Center of Chang Gung University under a 12-h light/dark cycle. Food and water were provided ad libitum. Lipopolysaccharide (LPS) is a vital component of the outer membrane in Gram-negative bacteria. LPS serves as the principal toxin derived from the gut microbiota, triggering inflammation in gut epithelial cells. Notably, the serum concentration of LPS is significantly elevated in patients with inflammatory bowel disease (IBD) [53], and LPS has been employed in experimental animal models to investigate intestinal injury [54]. In this study, WT and FXR-KO mice were intraperitoneally injected with a single dose of LPS (5 mg/kg) and after 12 h injected with GW4064 (20 mg/kg i.p. every 12 h) twice. After 6 h of the last GW4064 treatment, all animals were terminated by gradual fill isoflurane asphyxiation.

### 4.2. Histology

The colon tissues were fixed in 4% paraformaldehyde for 48 h and cut into 5 μm-thick serial sections, while hematoxylin and eosin (HE) staining was performed for histology. Then, the colon tissue damage was scored according to the histopatho-logical index (Appendix A) [55]. Slides were incubated with 7.5% non-immune goat serum for 30 min and then incubated with primary antibody for 2 h. Immunofluorescence (IF) and immunohistochemical (IHC) staining were performed as previously described [56]. The antibodies used for IF and IHC are listed in Appendix A.

### 4.3. Western Blot Analysis

The colon tissue protein extraction used NE-PER nuclear and cytoplasmic extraction reagents. The protein concentration was determined by the use of the Bio-Red Protein Assay Kit, in which protein extracts were added to the prepared SDS-PAGE for protein electrophoresis. Next, protein extracts were transferred onto PVDF membranes and blocked with 5% skim milk powder solution and then incubated with diluted primary antibodies overnight at 4 °C and incubated with HRP-conjugated secondary antibody for 1 h. The signals were detected using an enhanced chemiluminescence kit and quantified using ImageQuant 5.2 software.

### 4.4. Quantitative Reverse Transcription Polymerase Chain Reaction (qRT-PCR)

RNA was isolated from the colon of mice using TRIzol reagent, and cDNA was generated from 1 μg total RNA using the High-Capacity cDNA Reverse transcription kit. qRT-PCR analysis was carried out using SYBR green PCR mastermix and analyzed on a LightCycler^®^ 1.5 Real-Time PCR System. Values were normalized to glyceraldehyde-3-phosphate dehydrogenase (*Gapdh*). The sequences of primers used for qRT-PCR are listed in Appendix A.

### 4.5. 16S rRNA Gene Amplification and Phylogenetic Analysis

Fecal samples were collected and immediately transferred to a −80 °C freezer. Total bacterial DNA was extracted and purified using the QIAGEN mini stool kit (QIAGEN, Hilden, Germany). The total DNA was extracted with the reported methods [57]. The primers 16s_illumina_V3F (5′-CCTACGGGNGGCWGCAG-3′) and 16s_illumina_V4R (5′-GACTACHVGGGTATCTAATCC-3′) were used for bacterial 16S rRNA variable region V3-V4. The 16S rRNA sequences were compared for similarity with the reference species of bacteria using the *NCBI nucleotide database* and *PhiX Control Library*. To begin, sequences were clustered into operational taxonomic units (OTUs) using UPARSE [58]. To generate taxonomic assignments, Bowtie2 was used to align sequencing reads against the collection of a 16S rRNA sequences database [58]. A principal component analysis (PCA) plot was performed using ClustVis 2.0 software [59], which is a web tool for visualizing clustering of multivariate data using principal component analysis and heatmap from Oxford University.

### 4.6. Plasma, Liver, and Colonic Bile Acids Analysis

Plasma (50 uL) was added to 150 μL of methanol with internal standard deoxycholic-2,2,4,4,11,11-d6 acid (DCA-d6; 809675, Merck, Rahway, NJ, USA) for protein precipitation and remained in −80 °C for 10 min [60]. Samples were centrifuged for 30 min at 12,000× *g* at 4 °C. The frozen liver and colon tissue were weighed and placed in homogenization tubes containing ceramic beads. The tissues were homogenized in a Precelly24 homogenizer two times over 15 s at 6000 rpm with 30 s pause intervals with extraction solvent (150 μL) containing internal standard. After homogenization, samples were centrifuged for 30 min at 12,000× *g* at 4 °C. The supernatant was analyzed with UPLC/MS/MS [61].

### 4.7. Fecal SCFAs Extraction and Derivatization

Frozen fecal samples were thawed at room temperature, then 50 mg of fecal material was combined with 1 mL of 10% isobutanol (J.T. baker 9044-01). The mixture was homogenized for 20 s at 6000 rpm with two cycles and a 30 s pause in between, using a Precelly24 homogenizer (Bertin Technologies, Montigmy le Bretonnexux, France). Subsequently, the homogenized mixture was centrifuged for 30 min at 12,000 rpm at 4 °C. Following centrifugation, 270 µL of the supernatant was transferred into a fresh glass vial. Sequentially, 50 µL of 20 mM NaOH and 160 µL of chloroform were added to the same glass vial for extraction. After vortexing for 5 min, the sample was centrifuged for 15 min at 1200× *g*. The upper aqueous layer (200 µL) was then carefully transferred to a new glass vial for further derivatization. For both the calibration standards and the aqueous layer (200 µL), the following steps were taken: 70 µL of the internal standard (100 µM butyrate-d8), 80 µL of isobutanol, 100 µL of pyridine, and 200 µL of ultrapure water were added to the sample. Subsequently, 50 µL of isobutyl chloroformate was carefully introduced into the sample or standard solution. To release the gases generated by the reaction, the cap was left open for 20 s, then closed and mixed for 1 min. The mixture was then sonicated for 5 min. Finally, 150 µL of hexane was added and thoroughly mixed. The vials were centrifuged at 1200× *g* for 10 min. The upper hexane phase was transferred into an autosampler sample vial and securely capped for GC/MS analysis.

### 4.8. GC/MS Analysis

The GC/MS analysis was conducted using a Bruker Scion 436 gas chromatograph/mass spectrometry system (Bruker Daltonics, Billerica, MA, USA) equipped with a VF-5 ms capillary column (30 m in length, 0.25 mm in diameter, and a 0.5 µm film thickness) from Agilent Technologies, Santa Clara, CA, USA. Analyte quantification was performed in the selected ion monitoring (SIM) mode. The temperature settings for various components were as follows: the injector was set at 260 °C, the ion source at 250 °C, and the transfer line at 280 °C. The helium carrier gas flowed at a rate of 1 mL/min. A 2 µL volume of derivatized sample was injected with a split ratio of 50:1. The temperature program for the column was as follows: it started at 40 °C and was maintained for 5 min, then ramped to 200 °C at a rate of 10 °C/min, and held for 1 min. Finally, the temperature was increased to 310 °C at a rate of 50 °C/min and maintained at this level for 3 min.

### 4.9. Statistical Analysis

Results are presented as mean ± standard error of the mean (SEM). Statistical determined by the Student’s *t*-test and analysis of variance with Student’s Newman–Keuls multiple-range test as appropriate. All *p*-values values <0.05 were considered statistically significant.

## 5. Conclusions

This study elucidates the pivotal role of GW4064 in modulating FXR signaling, thereby reinforcing the significance of the FXR/αKlotho/βKlotho/FGFs pathway in gastrointestinal health. Notably, the observed reduction in tumor-promoting markers such as β-catenin and cyclin D1, particularly in the context of LPS-induced carcinogenesis, underscores the therapeutic potential of FXR-targeted approaches. In conclusion, the activation of FXR and the αKlotho/βKlotho/FGFs pathway may represent a novel approach to prevent intestinal epithelial barrier dysfunction and colon tumorigenesis.

## Figures and Tables

**Figure 1 ijms-24-16932-f001:**
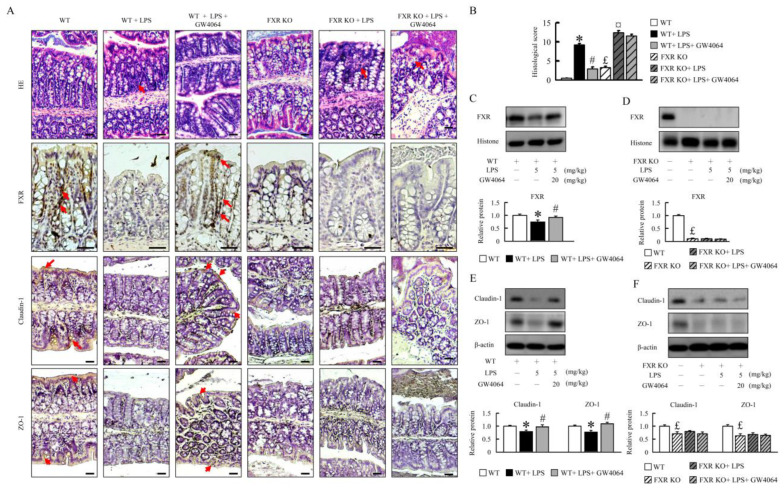
GW4064-dependent FXR activation ameliorates histological characteristics of tight junctions. (**A**) Representative hematoxylin and eosin staining and immunohistochemistry of FXR, claudin-1, and ZO-1 staining for colon sections are shown. Scale bar: 50 μm. One set of red arrows highlights tissues exhibiting inflammatory cell infiltration as observed in HE staining, while additional sets of red arrows denote areas demonstrating positive staining. (**B**) Histological score. The (**C**,**D**) FXR, (**E**,**F**) claudin-1, and ZO-1 protein were determined by Western blot. Values are expressed as the mean ± standard error of the mean (*n* = 5 mice per group). * *p* < 0.05, WT vs. WT + LPS; # *p* < 0.05, WT + LPS vs. WT + LPS + GW4064; £ *p* < 0.05, WT vs. FXR KO; ¤ *p* < 0.05, FXR KO vs. FXR KO + LPS.

**Figure 2 ijms-24-16932-f002:**
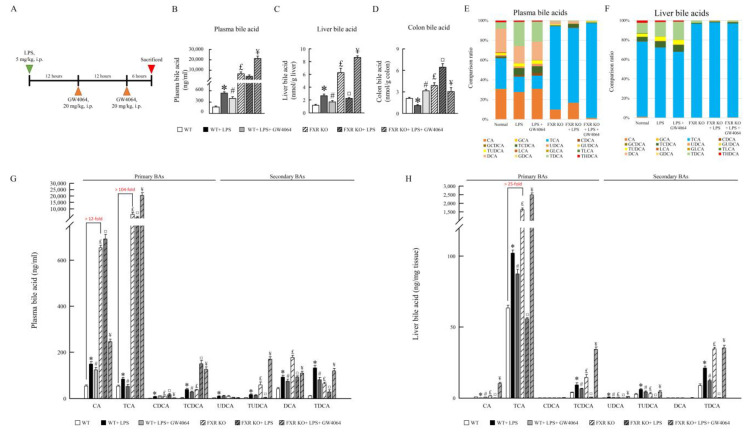
GW4064 affects bile acids profile. (**A**) Schematic diagram of the experiment. The concentration of (**B**) plasma, (**C**) liver, (**D**) colon bile acids, and (**E**–**H**) individual bile acids in LPS-treated wild-type (WT) and FXR-knockout (FXR-KO) mice with and without GW4064. Values are expressed as the mean ± standard error of the mean (*n* = 5 mice per group). * *p* < 0.05, WT vs. WT + LPS; # *p* < 0.05, WT + LPS vs. WT + LPS + GW4064; £ *p* < 0.05, WT vs. FXR KO; ¤ *p* < 0.05, FXR KO vs. FXR KO + LPS; ¥ *p* < 0.05, FXR KO + LPS vs. FXR KO + LPS + GW4064.

**Figure 3 ijms-24-16932-f003:**
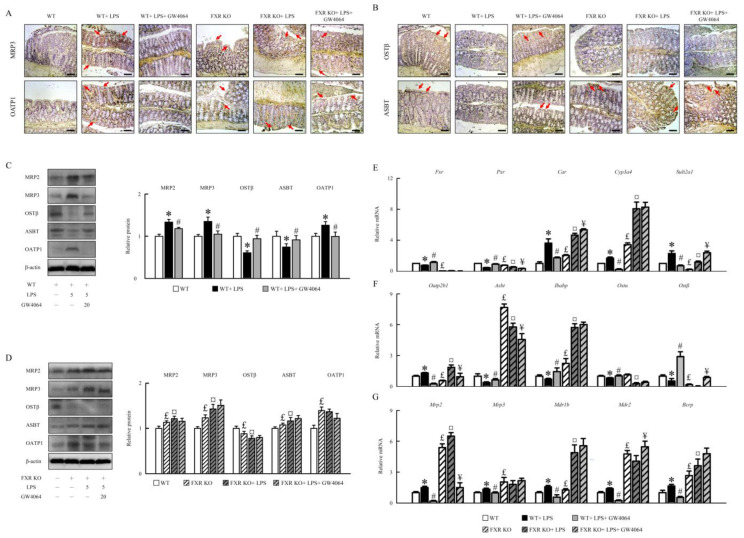
The effects of GW4064 on bile acid receptors and transporters expression. Immunohistochemistry of colon sections for (**A**) MRP3, OATP1, (**B**) OSTβ, and ASBT staining. Scale bar: 100 μm. Red arrows indicate the tissues with positive staining. (**C**,**D**) The MRP2, MRP3, OSTβ, ASBT, and OATP1 proteins were determined by Western blot. qRT-PCR analyses of colonic mRNA expression of (**E**) *Fxr*, *Pxr*, *Car*, *Cyp3a11*, and *Sult2a1*; (**F**) *Oatp2b1*, *Asbt*, *Ibabp*, *Ostα*, and *Ostβ*; and (**G**) *Mrp2*, *Mrp3*, *Mdr1b*, *Mdr2*, and *Bcrp*. Values are expressed as the mean ± standard error of the mean (*n* = 5 mice per group). * *p* < 0.05, WT vs. WT + LPS; # *p* < 0.05, WT + LPS vs. WT + LPS + GW4064; £ *p* < 0.05, WT vs. FXR KO; ¤ *p* < 0.05, FXR KO vs. FXR KO + LPS; ¥ *p* < 0.05, FXR KO + LPS vs. FXR KO + LPS + GW4064.

**Figure 4 ijms-24-16932-f004:**
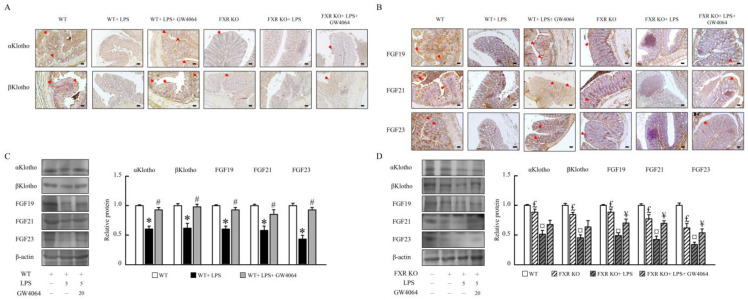
GW4064 regulates αKlotho /βKlotho /FGF19/FGF21/FGF23 pathway. Immunohistochemistry of colon sections for (**A**) αKlotho and βKlotho and (**B**) FGF19, FGF21, and FGF23 staining. Scale bar: 50 μm. Red arrows indicate the tissues with positive staining. The (**C**) αKlotho and βKlotho and (**D**) FGF19, FGF21, and FGF23 protein levels were determined by Western blot. * *p* < 0.05, WT vs. WT + LPS; # *p* < 0.05, WT + LPS vs. WT + LPS + GW4064; £ *p* < 0.05, WT vs. FXR KO; ¤ *p* < 0.05, FXR KO vs. FXR KO + LPS; ¥ *p* < 0.05, FXR KO + LPS vs. FXR KO + LPS + GW4064.

**Figure 5 ijms-24-16932-f005:**
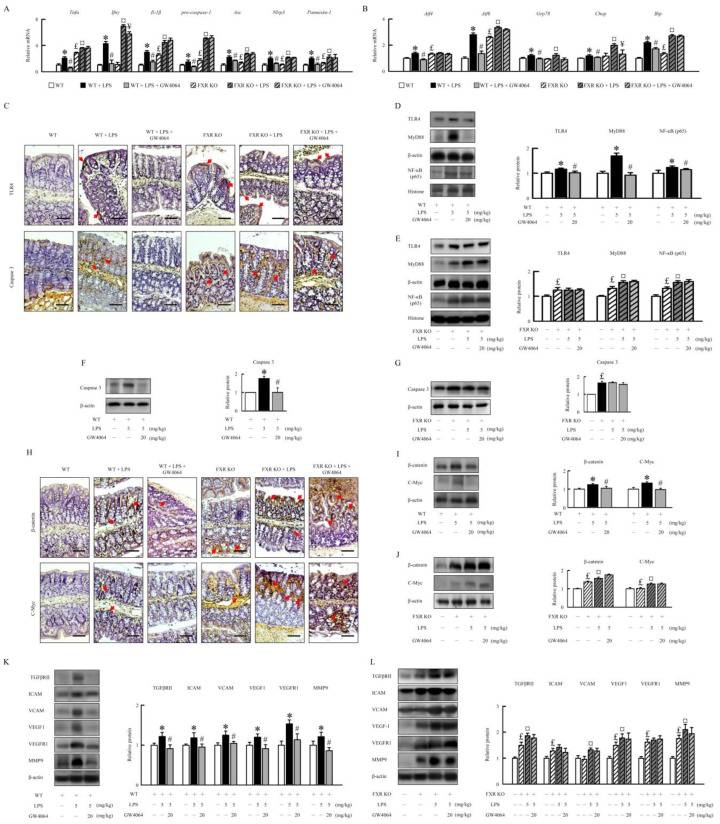
FXR deletion exhibits signs of TLR4-mediated inflammatory response and apoptosis of IECs and leads to ER stress. qRT-PCR analyses of colonic mRNA levels of (**A**) *Tnfα*, *Ifnγ*, *Il-1β*, *pro-caspase-1*, *Asc*, *Nlrp3*, and *pannexin-1* and (**B**) *Atf4*, *Atf6*, *Grp78*, *Chop*, and *Xbp1s*. (**C**) Immunohistochemistry of colon sections for TLR4 and caspase 3 staining. Scale bar: 100 μm. Red arrows indicate the positive staining tissues. (**D**,**E**) The TLR4, MyD88, and NF-κB protein were determined by Western blot. (**F**,**G**) The caspase 3 protein levels were determined by Western blot. (**H**) Immunohistochemistry of colon sections for β-catenin and c-Myc staining. Red arrows indicate the positive staining tissues. Scale bar: 100 μm. (**I**,**J**) The β-catenin and c-Myc and (**K**,**L**) TGFβRII, ICAM, VCAM, VEGF1, VEGFR1, and MMP9 protein levels were determined by Western blot. Values are expressed as the mean ± standard error of the mean (*n* = 5 mice per group). * *p* < 0.05, WT vs. WT + LPS; # *p* < 0.05, WT + LPS vs. WT + LPS + GW4064; £ *p* < 0.05, WT vs. FXR KO; ¤ *p* < 0.05, FXR KO vs. FXR KO + LPS; ¥ *p* < 0.05, FXR KO + LPS vs. FXR KO + LPS + GW4064.

**Figure 6 ijms-24-16932-f006:**
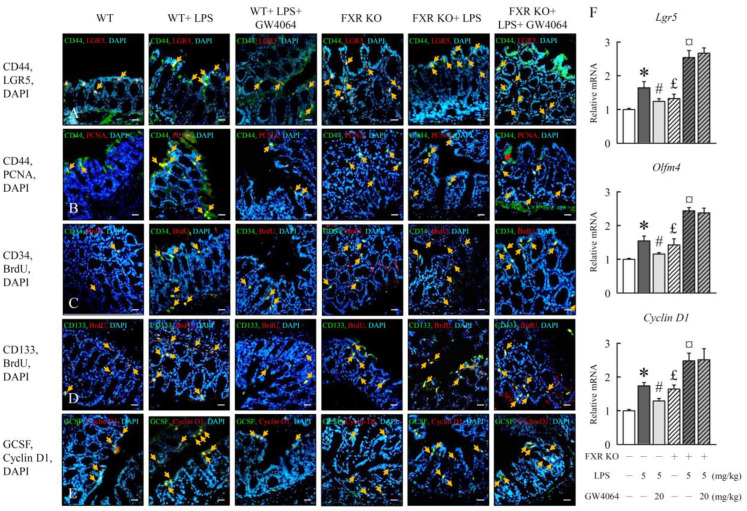
GW4064 alters colonic stem cell proliferation. (**A**) Double immunofluorescence staining of LGR5 (red) and CD44 (green); (**B**) PCNA (red) and CD44 (green); BrdU (red) and (**C**) CD34 (green); (**D**) BrdU (red) and CD133 (green); and (**E**) cyclin D1 (red) and GCSF (green) with DAPI (blue) in colon. Scale bar  =  50 μm. The yellow arrow signifies areas exhibiting positive outcomes in the double immunofluorescence staining procedure. (**F**) qRT-PCR analyses of colonic mRNA level of *Lgr5*, *Olfm4*, and *cyclin D1*. Values are expressed as the mean ± standard error of the mean (*n* = 5 mice per group). * *p* < 0.05, WT vs. WT + LPS; # *p* < 0.05, WT + LPS vs. WT + LPS + GW4064; £ *p* < 0.05, WT vs. FXR KO; ¤ *p* < 0.05, FXR KO vs. FXR KO + LPS.

**Figure 7 ijms-24-16932-f007:**
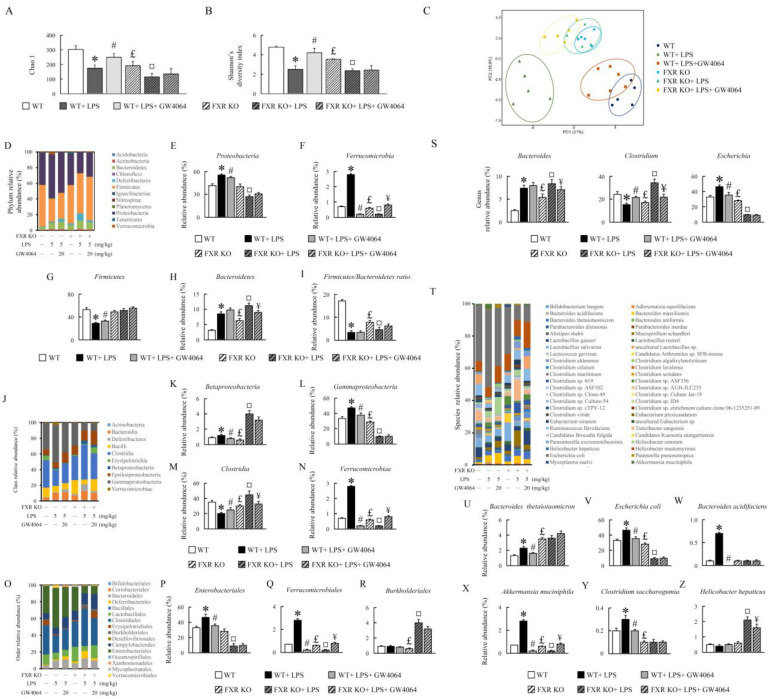
GW4064 alters intestinal microbial composition in mice. Alpha diversity measurements of the microbiota across locations. (**A**) Chao1 and (**B**) Shannon’s diversity index. PCA plot based on the abundance of bacterial gene sequences in fecal content. (**C**) Axes correspond to principal component 1 (*x* axis) and 2 (*y* axis). (**D**) Microbial community bar plot by phylum relative abundance (%). (**E**–**I**) Phylum levels of *Proteobacteria*, *Verrucomicrobia*, *Firmicutes*, *Bacteroidetes*, and *Firmicutes*/*Bacteroidetes* (F/B) ratio. (**J**) Microbial community bar plot by class relative abundance (%). (**K**–**N**) Class levels of *Betaproteobacteria*, *Gammaproteobacteria*, *Clostridia*, and *Verrucomicrobiae*. (**O**) Microbial community bar plot by order relative abundance (%). (**P**–**R**) Order levels of *Enterobacteriales*, *Verrucomicrobiales*, and *Burkholderiales*. (**S**) Genus levels of *Bacteroides*, *Clostridium*, and *Escherichia*. (**T**) Microbial community bar plot by species relative abundance (%). (**U**–**Z**) Species levels of *Bacteroides thetaiotaomicron*, *Escherichia coli*, *Bacteroides acidifaciens*, *Akkermansia muciniphila*, *Clostridium saccharogumia*, and *Helicobacter hepaticus*. Values are expressed as the mean ± standard error of the mean (*n* = 5 mice per group). * *p* < 0.05, WT vs. WT + LPS; # *p* < 0.05, WT + LPS vs. WT + LPS + GW4064; £ *p* < 0.05, WT vs. FXR KO; ¤ *p* < 0.05, FXR KO vs. FXR KO + LPS; ¥ *p* < 0.05, FXR KO + LPS vs. FXR KO + LPS + GW4064.

**Figure 8 ijms-24-16932-f008:**
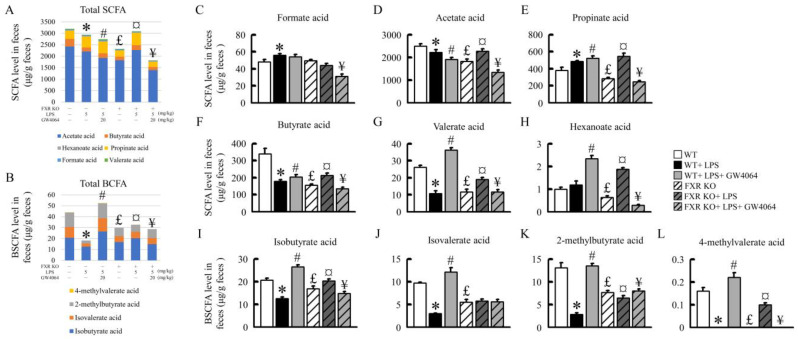
GW4064 alters SCFA and BCFA in feces. (**A**) Total SCFA and (**B**) total BCFA levels in feces. (**C**–**H**) Formate acid, acetate acid, propinate acid, butyrate acid, valerate acid, and hexanoate acid levels in feces. (**I**–**L**) Isobutyrate acid, isovalerate acid, 2-methylbutyrate acid, and 4-methylvalerate acid levels in feces. Values are expressed as the mean ± standard error of the mean (*n* = 5 mice per group). * *p* < 0.05, WT vs. WT + LPS; # *p* < 0.05, WT + LPS vs. WT + LPS + GW4064; £ *p* < 0.05, WT vs. FXR KO; ¤ *p* < 0.05, FXR KO vs. FXR KO + LPS; ¥ *p* < 0.05, FXR KO + LPS vs. FXR KO + LPS + GW4064.

**Figure 9 ijms-24-16932-f009:**
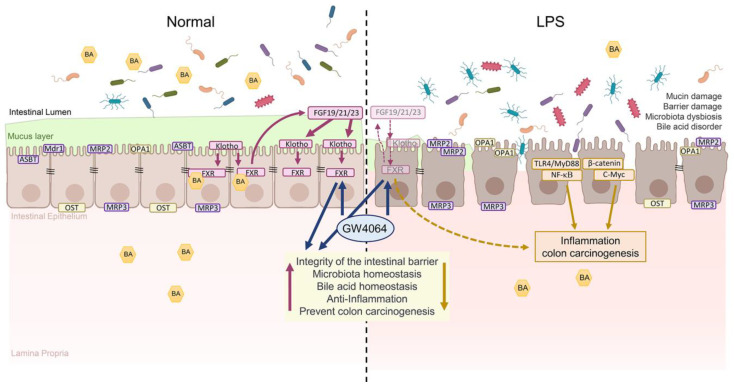
Representation of the inflammatory process induced in the colonic mucosa of LPS models. LPS causes disruption of the intestinal epithelial barrier and thereby enables the entry of luminal bacteria or bacterial antigens into the mucosa. Activation of the FXR/αKlotho/βKlotho/FGFs pathway may be a potential strategy for preventing intestinal barrier dysfunction and colon tumorigenesis.

## Data Availability

Data is contained within the article and Appendix A.

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
