# Peer review of "Farnesoid X Receptor Agonist GW4064 Protects Lipopolysaccharide-Induced Intestinal Epithelial Barrier Function and Colorectal Tumorigenesis Signaling through the αKlotho/βKlotho/FGFs Pathways in Mice"

_ijms, 2023, doi:10.3390/ijms242316932_

Round 1

Reviewer 1 Report

Comments and Suggestions for Authors

In this study, the authors investigated the role of the FXR/βKlotho/fibroblast growth factors (FGFs) pathway in maintaining intestinal barrier functions. To see the role of this pathway, they challenge the wild-type mice with LPS to disrupt barrier function and treat them with an FXR agonist. They also used FXR knockout mice to confirm the role of the pathways. Overall the data are sound. I have only 2 comments:

1.       Please quantify the cell infiltration after LPS treatment and GW4064 treatment.

2.       Please show the actual cancer transformation after LPS treatment. Just showing changes in CD44, LGR5, PCNA, CD34, BrdU, CD133, GCSF, and cyclin D1 might not be enough to conclude that FXR is a suitable target for intestinal cancer. I understand its role in inflammation.

Comments on the Quality of English Language

The quality of English language looks fine.

Author Response

Thank you for your valuable comments on our manuscript.

Comment 1:

"Please quantify the cell infiltration after LPS treatment and GW4064 treatment."

Response:

While direct cell quantification wasn’t conducted in our preliminary experiments, we propose a viable alternative methodology utilizing our existing dataset. Our strategy entails the implementation of histological scoring techniques to evaluate cell infiltration. This involved a meticulous analysis of our histological samples, focusing on variables such as the degree of inflammatory cell infiltrate, compromise of mucosal integrity, loss of crypt architecture, and the extent of pathological alterations, as depicted in Figure 1 B. Utilizing a standardized histological scoring framework will enable us to furnish a quantifiable evaluation of cell infiltration, linking seamlessly with our prior results. This approach is a well-established and robust method in gastrointestinal research, having been successfully applied in analogous studies for the quantification of inflammatory response metrics. We have also cited reference as the basis for this alternative approach. Asgharzadeh, F.; Yaghoubi, A.; Nazari, S.E.; Hashemzadeh, A.; Hasanian, S.M.; Avan, A.; Javandoost, A.; Ferns, G.A.; Soleimanpour, S.; Khazaei, M. The beneficial effect of combination therapy with sulfasalazine and valsartan in the treatment of ulcerative colitis. EXCLI journal 2021, 20, 236-247. (reference 55)

Comment 2:

"Please show the actual cancer transformation after LPS treatment."

Response: Building upon established scholarly consensus regarding the pivotal roles of CD44 and CD133 in colorectal cancer, our study leveraged fluctuations in the expression of CD44 and CD133 in the intestinal region as indicators of actual cancerous transformation. Our empirical findings indicated that normal mice exposed to LPS treatment showed a twofold increase in CD44 and CD133 expression levels in the colon compared to their untreated counterparts. In a parallel observation, FXR KO mice treated with LPS demonstrated a comparable twofold escalation in these markers' expression in the colon when contrasted with control FXR KO mice. Consequently, our research concluded that LPS treatment significantly amplified CD44 and CD133 expression in the colon in both normal and FXR KO mice, signifying potential markers of colorectal cancer transformation. (Lines 578-585; Ref. 41-44)

Reviewer 2 Report

Comments and Suggestions for Authors

Thank you for the opportunity to read this manuscript. The paper is very good.  

Minor comments : 

1) abbreviations - please spell out all abbreviations

2) citation should be in MDPI style.

3) I can't see SUMMARY of the results, I know that IJMS have different standard of the manuscript but please add maybe 2-3 sentences in SUMMARY it can be as SUBchapter 

4) If you have more interesting images please extend it in supplementary material

5) what kind of study ?  please specify type of study in the title (in vitro or whatever you think is but please specify!)

In my opinion to much GGTGCGT and so on. This in table II is useless. Please send it to supplementary not in main document.

Regards

Author Response

Response to Reviewer 2,

Response to Comment 1:

Abbreviations: We ensure that all abbreviations are spelled out in their first instance to enhance clarity and readability (Supplementary Table S4).

Response to Comment 2:

Citation Style: All citations are revised to adhere to the MDPI citation style, as per the journal's guidelines.

Response to Comment 3:

I can't see SUMMARY of the results, I know that IJMS have different standard of the manuscript but please add maybe 2-3 sentences in SUMMARY it can be as SUBchapter.

A concise summary of the results has been included as a subchapter to encapsulate the primary findings of our study in the revised manuscript. (Lines 441-448)

Response to Comment 4:

If you have more interesting images please extend it in supplementary material.

We have carefully considered your recommendation and appreciate your interest in further enriching the visual content of our paper. However, after thorough deliberation, we have opted not to extend the supplementary material with additional images at this point in time. Our decision is grounded in the belief that our current dataset and visuals adequately and comprehensively represent the full scope and findings of our study. In the interest of maintaining the clarity and focus of our presentation, we have chosen to prioritize the quality and relevance of the existing images. Your expertise is invaluable to us, and we are committed to making adjustments that enhance the overall quality of our manuscript based on your feedback.

Response to Comment 5:

what kind of study? please specify type of study in the title (in vitro or whatever you think is but please specify!):

We have already revised the title to reflect that the study was conducted in mice. We have revised title as “Farnesoid X receptor agonist GW4064 protects lipopolysaccharide-induced intestinal epithelial barrier function and colorectal tumorigenesis signaling through the αKloth/βKlotho/FGFs pathways in mice.” (Please see title page)

Response to Comment 6:

In my opinion to much GGTGCGT and so on. This in table II is useless. Please send it to supplementary not in main document.

Regarding your suggestion about the excessive use of gene sequences (e.g., GGTGCGT) in Table II, we agree with your assessment. To maintain the focus and readability of the main document, we have relocated these details to the Supplementary Table S3.

Reviewer 3 Report

Comments and Suggestions for Authors

This article about investigating the role of FXR/Klothos/FGFs pathways in 24 lipopolysaccharide (LPS)-induced intestinal barrier dysfunction and colon carcinogenesis is well-written, organized, and worth pursuing. It is easy to read and understand. 

As to originality and scientific sanity – I think this is acceptable.

I have, though, some suggestions:

1.  In the text, the reference numbers must be placed at the end of the sentence( ex. in lines 45-46, 524-525, and so on).

2. My conclusion is that activating FXR and the αKloth/βKlotho/FGFs pathway might be a new strategy for preventing intestinal epithelial barrier dysfunction and colon tumorigenesis.

But the conclusions section is missing which puts the readers in a difficult situation. Please make a conclusion with a future perspective. 

Comments on the Quality of English Language

Minor editing of English language required. 

Author Response

Response to Reviewer 3,

Response to Comment 1:

Placement of Reference Numbers: We have ensured that all reference numbers are correctly placed at the end of sentences, as you have advised. This will be meticulously applied throughout the manuscript, particularly in the lines you've highlighted.

Response to Comment 2:

Conclusion Section: The comprehensive conclusion section that not only summarizes our findings but also offers a future perspective. A detailed conclusion with a clear the study's implications have added in the revised manuscript. (Lines 815-821)

Response to Comment 3:

Regarding the quality of English language, we have taken significant revised to enhance the quality of the English language in our manuscript, as indicated in your previous feedback. We have engaged the services of a proficient English language colleague who meticulously proofread and edited the entire paper and ensuring that the revised manuscript meets the highest standards of academic writing.

Round 2

Reviewer 3 Report

Comments and Suggestions for Authors

Accept in present form.